# Uremic pruritus and long-term morbidities in the dialysis population

**Sze-Wen Ting**[1], **Pei-Chun Fan**[2], **Yu-Sheng Lin**[3,4], **Ming-Shyan Lin**[3], **Cheng-Chia Lee**[2], **George Kuo**[2], **Chih-Hsiang Chang**[2,4]*

**1** Department of Dermatology, New Taipei City Tu-Cheng Municipal Hospital, New Taipei City (Built and Operated by Chang Gung Medical Foundation), Taoyuan, Taiwan, **2** Department of Nephrology, Kidney Research Center, Linkou Chang Gung Memorial Hospital, Taoyuan, Taiwan, **3** Department of Cardiology, Chiayi Chang Gung Memorial Hospital, Puzi City, Taiwan, **4** Chang Gung University, College of Medicine, Taoyuan, Taiwan

* sunchang@cgmh.org.tw, franwisandsun@gmail.com

**Data Availability Statement:** Data Availability Statement: The data underlying this study is from the National Health Insurance Research Database (NHIRD), which has been transferred to the Health and Welfare Data Science Center (HWDC). The

## Abstract

### Background

Uremic pruritus (UP) is a multifactorial problem that contributes to low quality of life in dialysis patients. The long-term influences of UP on dialysis patients are still poorly understood. This study aims to elucidate the contribution of UP to long-term outcomes.

### Materials and method

We used the Taiwan National Health Insurance Research Database to conduct this study. Patients on chronic dialysis were included and divided into UP and non-UP groups according to the long-term prescription of antihistamine in the absence of other indications. The outcomes include infection-related hospitalization, catheter-related infection, major adverse cardiac and cerebrovascular events (MACCE) and parathyroidectomy.

### Results

After propensity score matching, 14,760 patients with UP and 29,520 patients without UP were eligible for analysis. After a mean follow-up of 5 years, we found that infection-related hospitalization, MACCE, catheter-related infection, heart failure and parathyroidectomy were all slightly higher in the UP than non-UP group (hazard ratio: 1.18 [1.16–1.21], 1.05 [1.01–1.09], 1.16 [1.12–1.21], 1.08 [1.01–1.16] and 1.10 [1.01–1.20], respectively). Subgroup analysis revealed that the increased risk of adverse events by UP was generally more apparent in younger patients and patients who underwent peritoneal dialysis.

### Conclusion

UP may be significantly associated with an increased risk of long-term morbidities.

## Introduction

Uremic pruritus (UP) is a common problem among patients with advanced chronic kidney disease (CKD) and end-stage renal disease (ESRD) receiving chronic dialysis [1–3]. The

NHIRD is not free to public access, and therefore interested researchers can obtain the data through formal application to the HWDC, Department of Statistics, Ministry of Health and Welfare, Taiwan (http://dep.mohw.gov.tw/DOS/np-2497-113.html). The authors had no special access privileges that others would not have.

**Funding:** This work was supported by Taiwan Ministry of Science and Technology in the form of a grant awarded to CHC (106-2314-B-182A-118-MY3) and Taiwan Clinical Trial Consortium in the form of grants awarded to CHC (MOST 106-2321-B-182-002, 105-2314-B-002-045) The funders had no role in study design, data collection and analysis, decision to publish, or preparation of the manuscript.

**Competing interests:** The authors have declared that no competing interests exist.

**Abbreviations:** UP, Uremic pruritus; MACCE, Major adverse cardiac and cerebrovascular events; ESRD, End-stage renal disease; NHIRD, National Health Insurance Research Database; ICD-9-CM, International Classification of Diseases, Ninth Revision, Clinical Modification diagnostic codes; SDH, Subdistribution hazard model; HR, Hazard ratio; SHR, Subdistribution hazard ratio; PD, Peritoneal dialysis; HD, Hemodialysis; PTH, Parathyroid hormone; IDWG, Interdialytic weight gain; BNP, Brain-type natriuretic peptide; TRP, Transient receptor potential channel; TRPV1, Transient receptor potential vanilloid 1; TRPV3, Transient receptor potential vanilloid 3; VAS, Visual analogue scale.

mechanism of UP is not fully understood, and current evidence suggests that its pathogenesis is multifactorial. The uremic toxin and metabolite hypothesis are supported by the observation that secondary hyperparathyroidism, hyperphosphatemia, hypercalcemia, elevated β-2 microglobulin, and inadequate dialysis clearance are associated with pruritus [3–6]. Local inflammatory mediators such as histamine and interleukin-31 may also play roles in pruritogenesis [7–9]. The opioid receptor has garnered attention because the k-opioid receptors agonist has an excellent effect on modulating pruritus in clinical trials [10, 11]. Substance P may also participate in pruritogenesis [12]. In addition, systemic therapy with gamma-aminobutyric acidergic agonist gabapentin and pregabalin also effective [1]. These all indicate that the pathogenesis of UP involves the interplay of uremic toxins, local inflammation, and neural circuit alterations.

UP not only negatively affects quality of life, sleep, and mood but may also contribute to worse long-term outcomes [2–4]. Our group reported that UP is associated with increased all-cause mortality, cardiovascular (CV) death and infection-related death [13]. In the current study, we further investigate the influence of UP on various types of long-term morbidities after starting dialysis therapy and also performed subgroup analysis on prespecified subgroups in the dialysis population by using Taiwan's nationwide population-based database.

## Materials and method

### Data source and ethic statement

For this study, data were analyzed from the National Health Insurance Research Database (NHIRD) in Taiwan, which is a nationwide research database containing no identifiable personal information. The NHIRD contains 99.8% coverage for the 23 million residents in Taiwan and provides all data of inpatient and outpatient services, diagnosis, prescriptions, examinations, operations, and expenditures. Further information regarding NHI and NHIRD have been described in previous publications [14–16]. Because the NHIRD contains no identifiable personal information, the need for informed consent was waived because of its confidentiality and privacy are well-maintained. The study was approved by the Institutional Review Board of Chang Gung Memorial Hospital.

### Study population and definition of uremic pruritus

The study population comprised patients who had been diagnosed with ESRD and were receiving permanent dialysis between January 1, 2001 and December 31, 2013. Permanent dialysis was verified by possession of a catastrophic illness certificate with the International Classification of Diseases, 9th Revision, Clinical Modification (ICD-9-CM) diagnostic code 585. The date of application for the catastrophic illness certificate was defined as the index date.

We excluded patients with missing demographics, who were younger than 20 years old, who received parathyroidectomy, or who survived for less than 180 days after dialysis initiation to identify the population with stable dialysis. Previous studies defined a meaningful pruritus occur more than three times within a 2-week period or with a lower frequency but last longer than 6 months. Studies using NHIRD defined patients with a 42 or higher daily doses of anti-histamine to have uremic pruritus. By combining these tow criteria, we defined UP as the prescription of 84 daily doses of antihistamines or more, or received ultraviolet B phototherapy within 1 year after dialysis initiation. On the other hand, patients who received less than 42 daily doses of antihistamines were classified as non-UP. Patients who have diagnoses that requiring long-term use of antihistamine, including allergic rhinitis (ICD-9-CM 477.xx), urticaria (ICD-9-CM 708), psoriasis (ICD-9-CM 696), mycosis fungoides (ICD-9-CM 202.1), or Sezary disease (ICD-9-CM 202.2), are classified into non-UP group [17–21]. Patients who

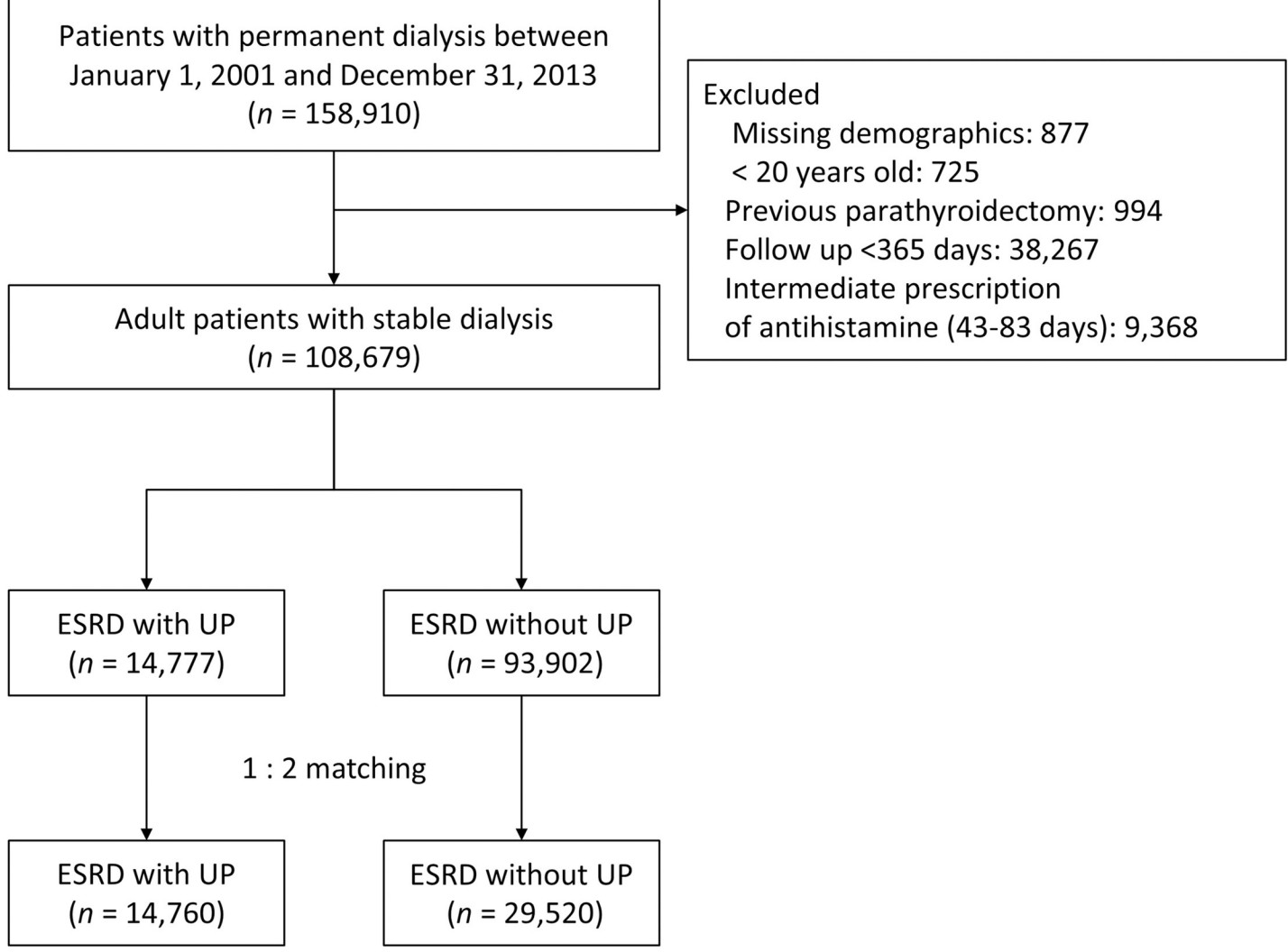

**Fig 1. Flowchart of study patient inclusion.**

received 43 to 83 daily doses of anti-histamine within 1 year of dialysis were excluded because the antihistamine doses were inadequate to classify a "frequent and troublesome pruritus" [19–21] (Fig 1). The pruritus usually occurs at the stage of advanced CKD, and some continued after staring dialysis. Because our main interest was the pruritus that persists after starting dialysis, those patients who received antihistamine before index date were not excluded.

## Definitions of covariates

Disease was identified using ICD-9-CM diagnostic codes (S1 Table). Covariates included age, sex, initial dialysis modality, dialysis access type for hemodialysis, 10 comorbidities and history of events, and 14 types of medication. Comorbidity was defined as receipt of 2 outpatient diagnoses or one inpatient diagnosis in the previous year. Event history (including stroke, heart failure, and myocardial infarction) was determined using any inpatient diagnosis before the index date, with diagnoses available from the year 1997 onwards. Many of the diagnoses of these diseases in the NHIRD have been validated in other studies [22, 23]. Medication

information was obtained using the claims data of outpatient visits or from pharmacy refills within 3 months prior to or after the index date.

### Definitions of outcomes

The outcomes of this study were infection-related hospitalization, major adverse cardiac and cerebrovascular events (MACCE), catheter-related infection, and parathyroidectomy. The MACCE in our study includes acute myocardial infarction, acute ischemic stroke, intracerebral hemorrhage, and heart failure. We also performed subgroup analysis on these outcomes, including infection-related hospitalization, MACCE, catheter-related hospitalization, heart failure, and parathyroidectomy.

Death was defined as withdrawal from the National Health Insurance program. The diagnostic codes of components of MACCE have been validated in other studies [24–28]. The incidences of MACCE were defined according to the principal inpatient diagnosis. Parathyroidectomy was extracted using the Taiwan National Health Insurance reimbursement codes of inpatient claims. Infection-related hospitalization was defined using principal or secondary discharge diagnoses, including bacteremia (candidemia, viremia and sepsis), cardiovascular, central nervous system, gastrointestinal, hepatobiliary and peritoneal infections, genitourinary, pulmonary, skin and soft tissue, bone and joint, dialysis access and central venous catheters, other device/procedure and surgery-related infection. All patients were followed from the index date to the date of event occurrence, date of death, or December 31, 2013, whichever came first.

### Statistical analysis

To reduce influence from potential confounding factors in this observational study, propensity score matching (PSM) was performed. The propensity score was the predicted probability of being in the UP group given the values of covariates according to the logistic regression. The variables selected for calculating propensity scores are listed in Table 1, and the follow-up year was replaced with the index date (Table 1). Each patient in the UP group was matched with 2 corresponding patients in the non-UP group. The matching was processed using a greedy nearest neighbor algorithm with a caliper of 0.2 times the standard deviation of the logit of propensity score, with random matching order, and without replacement. The quality of matching was evaluated using the absolute value of standardized difference (STD) between the groups, in which a value less than 0.1 was considered a negligible difference.

The incidences of outcomes between groups were compared using the Fine and Gray subdistribution hazard model, which considered death to be a competing risk. The study group (UP vs. non-UP) was the only explanatory variable in the survival analyses. The within-pair clustering of outcomes after PSM was accounted for using a robust standard error, which is known as a marginal model [29] Finally, subgroup analysis was performed to determine whether the effect of UP on several outcomes (which were significantly different between the UP and non-UP groups) was similar among different levels of the 5 prespecified subgroup variables, including age, sex, initial dialysis modality, diabetes, and ischemic heart disease.

A 2-sided P value < 0.05 was considered to be statistically significant, and no adjustment for multiple testing (multiplicity) was made in this study. All statistical analyses were performed using SAS version 9.4 (SAS Institute, Cary, NC, USA).

## Results

During the study period, we identified 158,910 patients who were diagnosed with ESRD and received long-term dialysis therapy. After exclusion of those patients with incomplete data

**Table 1. Baseline characteristics of dialytic patients with and without uremic pruritus.**

| Variable | Before matching | | | After matching | | |
| --- | --- | --- | --- | --- | --- | --- |
| | UP | Non-UP | STD | UP | Non-UP | STD |
| | (*n* = 14,777) | (*n* = 93,902) | | (*n* = 14,760) | (*n* = 25,920) | |
| Demographic | | | | | | |
| Age (years) | 62.7 ± 13.2 | 60.4 ± 14.2 | 0.17 | 62.6 ± 13.2 | 62.9 ± 13.5 | -0.02 |
| Age ≥65 years | 6,897 (46.7) | 38,301 (40.8) | 0.12 | 6,885 (46.6) | 13,850 (46.9) | -0.01 |
| Female | 7,321 (49.5) | 47,171 (50.2) | -0.01 | 7,311 (49.5) | 14,552 (49.3) | <0.01 |
| Initial modality | | | | | | |
| Hemodialysis (HD) | 12,791 (86.6) | 84,059 (89.5) | -0.09 | 12,776 (86.6) | 25,476 (86.3) | 0.01 |
| Peritoneal dialysis | 1,986 (13.4) | 9,843 (10.5) | 0.09 | 1,984 (13.4) | 4,044 (13.7) | -0.01 |
| Dialysis access type for hemodialysis | | | | | | |
| Fistula | 10,239 (69.3) | 71,362 (76.0) | -0.15 | 10,233 (69.3) | 20,373 (69.0) | 0.01 |
| Graft | 878 (5.9) | 4,375 (4.7) | 0.06 | 873 (5.9) | 1,759 (6.0) | <0.01 |
| Tunnel-catheter | 1,674 (11.3) | 8,322 (8.9) | 0.08 | 1,670 (11.3) | 3,344 (11.3) | <0.01 |
| Comorbidity | | | | | | |
| Polycystic kidney disease | 276 (1.9) | 1,814 (1.9) | <0.01 | 276 (1.9) | 565 (1.9) | <0.01 |
| Old dementia | 350 (2.4) | 1,997 (2.1) | 0.02 | 350 (2.4) | 700 (2.4) | <0.01 |
| Hypertension | 12,191 (82.5) | 69,932 (74.5) | 0.20 | 12,174 (82.5) | 24,772 (83.9) | -0.04 |
| Diabetes mellitus | 8,110 (54.9) | 43,372 (46.2) | 0.17 | 8,096 (54.9) | 16,470 (55.8) | -0.02 |
| Chronic obstructive pulmonary disease | 1,108 (7.5) | 6,118 (6.5) | 0.04 | 1,105 (7.5) | 2,236 (7.6) | <0.01 |
| Peripheral arterial disease | 466 (3.2) | 2,599 (2.8) | 0.02 | 466 (3.2) | 909 (3.1) | <0.01 |
| Ischemic heart disease | 3,967 (26.8) | 21,289 (22.7) | 0.10 | 3,964 (26.9) | 7,967 (27.0) | <0.01 |
| History of event | | | | | | |
| Old stroke | 2,506 (17.0) | 13,605 (14.5) | 0.07 | 2,499 (16.9) | 5,081 (17.2) | -0.01 |
| History of heart failure | 3,489 (23.6) | 18,923 (20.2) | 0.08 | 3,485 (23.6) | 6,897 (23.4) | 0.01 |
| Old myocardial infarction | 864 (5.8) | 4,447 (4.7) | 0.05 | 861 (5.8) | 1,698 (5.8) | <0.01 |
| Medications | | | | | | |
| Steroid | 1,625 (11.0) | 6,118 (6.5) | 0.16 | 1,610 (10.9) | 3,046 (10.3) | 0.02 |
| Other immunosuppressive agent | 250 (1.7) | 1,668 (1.8) | -0.01 | 250 (1.7) | 485 (1.6) | <0.01 |
| Antiplatelet | 4,269 (28.9) | 20,711 (22.1) | 0.16 | 4,261 (28.9) | 8,606 (29.2) | -0.01 |
| ACEI / ARB | 6,836 (46.3) | 35,237 (37.5) | 0.18 | 6,822 (46.2) | 13,870 (47.0) | -0.02 |
| Beta-blocker | 7,064 (47.8) | 36,842 (39.2) | 0.17 | 7,053 (47.8) | 14,427 (48.9) | -0.02 |
| Loop diuretics | 7,959 (53.9) | 40,032 (42.6) | 0.23 | 7,947 (53.8) | 16,170 (54.8) | -0.02 |
| K-sparing diuretics | 317 (2.1) | 1,394 (1.5) | 0.05 | 315 (2.1) | 625 (2.1) | <0.01 |
| Oral hypoglycemic agent | 4,699 (31.8) | 23,190 (24.7) | 0.16 | 4,690 (31.8) | 9,561 (32.4) | -0.01 |
| Insulin | 3,158 (21.4) | 15,990 (17.0) | 0.11 | 3,151 (21.3) | 6,349 (21.5) | <0.01 |
| Proton pump inhibitor | 2,587 (17.5) | 11,268 (12.0) | 0.16 | 2,578 (17.5) | 5,067 (17.2) | 0.01 |
| NSAID (including COX2) | 2,845 (19.3) | 9,882 (10.5) | 0.25 | 2,828 (19.2) | 5,320 (18.0) | 0.03 |
| Statin | 3,285 (22.2) | 16,661 (17.7) | 0.11 | 3,283 (22.2) | 6,661 (22.6) | -0.01 |
| Fibrate or Gemfibrozil | 848 (5.7) | 3,672 (3.9) | 0.09 | 845 (5.7) | 1,660 (5.6) | <0.01 |
| Vitamin D therapy | 1,992 (13.5) | 8,414 (9.0) | 0.14 | 1,981 (13.4) | 3,838 (13.0) | 0.01 |
| Follow-up duration (years) | 4.7 ± 3.1 | 5.6 ± 3.5 | -0.26 | 4.7 ± 3.1 | 4.9 ± 3.2 | -0.03 |

UP, uremic pruritus; STD, standardized difference; ACEI, angiotensin converting enzyme inhibitor; ARB, angiotensin receptor blocker; NSAID, non-steroidal anti-inflammatory drug; COX-2, cyclo-oxygenase-2 inhibitor

Data were presented as frequency (percentage) or mean ± standard deviation.

(n = 877), patients younger than 20 years of age (n = 725), patients who had received parathyroidectomy (n = 994), and patients for whom the follow-up duration was less than 365 days (n = 38,267), and patients who did not meet the criteria for anti-histamine dose (n = 9,368), we identified 14,777 dialysis patients in the UP group and 93,902 patients in the non-UP group (Fig 1). Before matching, the UP group were older, tend to use PD as the initial dialysis modality, less fistulas on hemodialysis initiation, had a higher prevalence of hypertension and diabetes, and received more medications (ASTD ≥ 0.1). After matching, all demographics, comorbidities, and medications were well-balanced between the groups (ASTD < 0.1; Table 1).

With a mean follow-up of 5 years (standard deviation: 3.3 years), the long-term outcomes are summarized in Table 2. The UP group exhibited a higher risk of long-term morbidities, including infection-related hospitalization (Subdistribution hazard ratio [SHR]: 1.13, 95% confidence interval [CI]: 1.16–1.21), catheter-related infection (SHR: 1.16, 95% CI: 1.12–1.21), MACCE (SHR: 1.05, 95% CI: 1.01–1.09), heart failure (SHR: 1.08, 95% CI: 1.01–1.16), and parathyroidectomy (SHR: 1.10, 95% CI: 1.01–1.20). Acute myocardial infarction, acute ischemic stroke, and intracerebral hemorrhage did not achieve statistical difference between the groups. The cumulative incidence functions of infection-related hospitalization and MACCE are depicted in Fig 2A and 2B.

As to the subgroup analysis of outcomes (which were significantly different between the UP and non-UP groups), the increased risk of infection-related hospitalization, MACCE, catheter-related infection, heart failure and parathyroidectomy by UP were mostly not different between age groups, sex, dialysis modality, presence of DM or ischemic heart disease, or use of vitamin D analogue (Fig 3A–3C, S1 and S2 Figs) The only significant interaction was found that the use of vitamin D analogue mitigated the detrimental effect of UP on catheter-related infection.

## Discussion

UP is not an uncommon problem among dialysis patients and impairs quality of life [2–4]. In our previous study, UP is associated with increased all-cause mortality, infection-related death, and CV death [13]. Pisoni et al. reported an increased mortality risk with severe pruritus, but this risk diminished after adjusting for sleep quality [3]. Narita et al. identified severe

**Table 2. Follow up outcome of in dialytic patients with and without uremic pruritus.**

| Outcome | Number of event (%) | | UP vs. non-UP | |
| --- | --- | --- | --- | --- |
| | UP | Non-UP | HR or SHR (95% CI) | *P*-value |
| | (*n* = 14,760) | (*n* = 29,520) | | |
| IRH | 10,110 (68.5) | 18,516 (62.7) | 1.18 (1.16–1.21) | <0.001 |
| Catheter related infection | 3,752 (25.4) | 6,580 (22.3) | 1.16 (1.12–1.21) | <0.001 |
| MACCE* | 3,599 (24.4) | 6,896 (23.4) | 1.05 (1.01–1.09) | 0.018 |
| Acute myocardial infarction | 947 (6.4) | 1,977 (6.7) | 0.95 (0.88–1.03) | 0.208 |
| Acute ischemic stroke | 1,543 (10.5) | 2,962 (10.0) | 1.04 (0.98–1.11) | 0.189 |
| Intracerebral hemorrhage | 451 (3.1) | 927 (3.1) | 0.97 (0.86–1.08) | 0.556 |
| Heart failure | 1,207 (8.2) | 2,236 (7.6) | 1.08 (1.01–1.16) | 0.035 |
| Parathyroidectomy | 789 (5.3) | 1,429 (4.8) | 1.10 (1.01–1.20) | 0.026 |

UP, uremic pruritus; HR, hazard ratio; SHR, subdistribution hazard ratio; CI, confidence interval; IRH, infection-related hospitalization; MACCE, major cardiovascular and cerebral composite adverse event

* Anyone of acute myocardial infarction, acute ischemic stroke, intracerebral hemorrhage, heart failure

Data were presented as frequency (percentage).

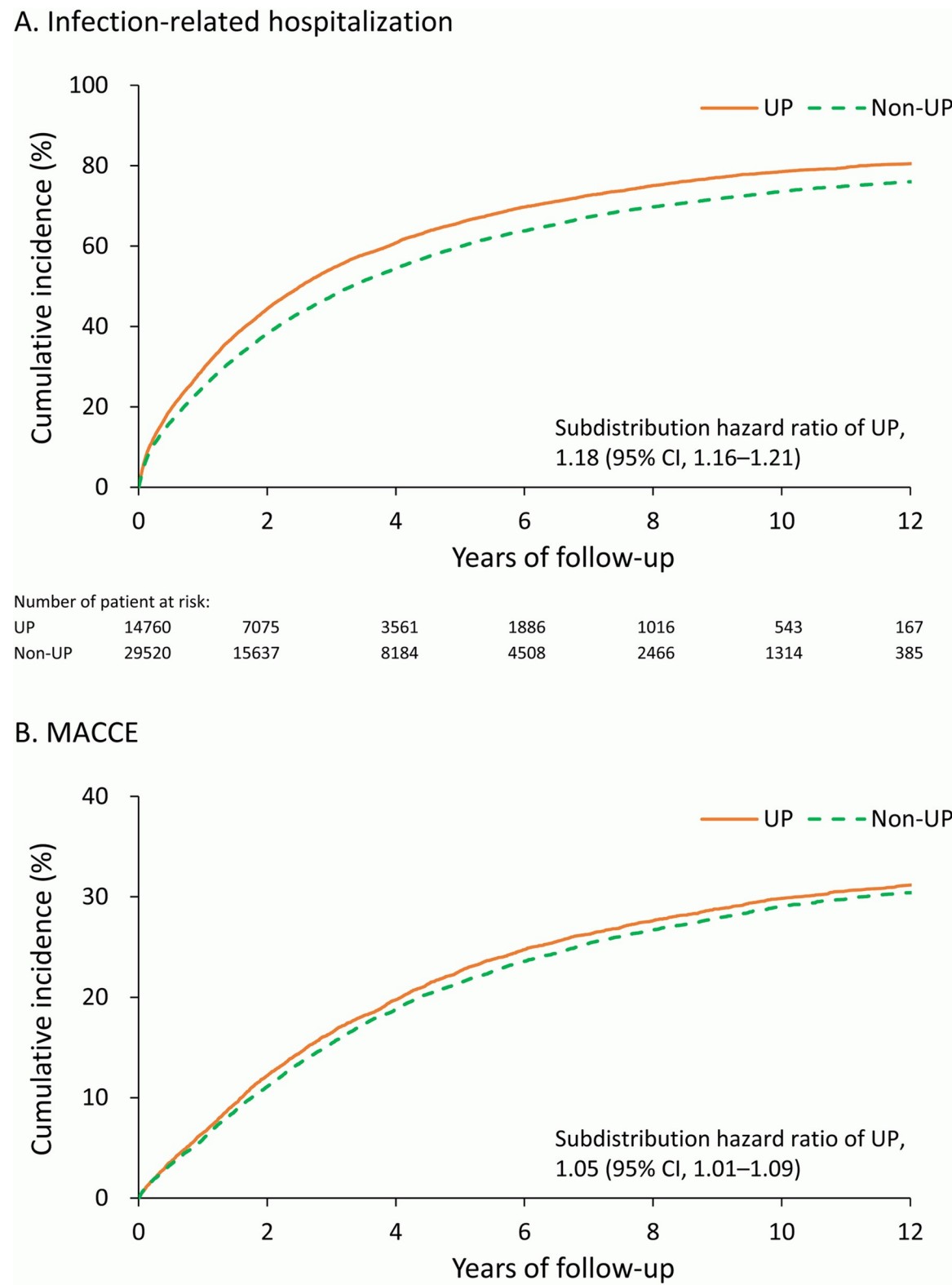

**Fig 2.** Unadjusted cumulative incidence function of infection-related hospitalization (A) and major cardiovascular and cerebral composite adverse event (B) in the propensity score matched cohort.

pruritus as a predictor of mortality, but they did not note differences among causes of death [2]. In 2014, Kimata et al. observed increased mortality among patients experiencing severe pruritus [4].

The association between UP and adverse long-term outcomes could be explained by several potential causes. Clinical studies have indicated that high PTH levels, hyperphosphatemia, and high urea nitrogen levels increase the intensity of pruritus [2, 3]. These factors usually indicate inadequate dialysis clearance or noncompliant dietary control. The elevation of PTH levels is also associated with left ventricular hypertrophy and CV events [30–33]. Several studies have reported amelioration of pruritus after parathyroidectomy in patients with uncontrolled secondary hyperparathyroidism undergoing dialysis [31, 34]. These reports correspond with our findings that UP is associated with higher mortality and increased rate of parathyroidectomy. The effect of UP on patients' outcomes were relatively small but significant. This might be explained that other traditional cardiovascular risk factors, including diabetes, hypertension, dyslipidemia and dialysis-specific risk factors, including hyperparathyroidism, deranged calcium-phosphate balance, all contributes to worse outcome in dialysis population. The presence of UP could be view as one comorbidity among them.

Noncompliant dietary control could result in hyperphosphatemia, hyperparathyroidism and excessive interdialytic weight gain (IDWG). Excessive IDWG usually causes fluid overload, left ventricular hypertrophy, hypertension, and CV events [35–37]. Brain-type natriuretic peptide (BNP) is a surrogate marker of volume overload; in addition, BNP and its receptor have also been found in transient receptor potential (TRP) vanilloid 1 (TRPV1)–expressing neurons [38, 39]. The TRPV1-expressing neuron transmit itch sensation from the peripheral C-fiber to the central nervous system [40]. Therefore, the increase of BNP with volume overload may have a role in itching transmission. Clinically, Shimizu et al. reported an association between elevated BNP and the severity of pruritus [41]. This provides a possible link between pruritus and volume overload in dialysis patients.

We found that the infection-related hospitalization was more frequent in the UP group. This corresponds with the study by Narita et al., in which more severe pruritus is associated with higher infection-related mortality [2]. A direct relationship between UP and infection has not yet been confirmed. We hypothesize that frequent scratching results in microtrauma of cutaneous barriers, which is supported by a systematic review demonstrating that exfoliating skin disease is a risk factor for cellulitis [42]. Systemically, the uremic milieu is associated with impaired immunity [43]. The presence of UP is frequently associated with increased uremic toxins and, consequently, worse immunity. Secondary hyperparathyroidism may mediate immune dysfunction in UP patients. Tzanno-Martins et al. reported that high PTH levels can impair antigen-stimulated T cell proliferation from dialysis patients and the inhibition disappeared after parathyroidectomy [44].

The subgroup analysis found a protective effect from vitamin D analogue in patients with UP, and this diminished the excessive catheter-related infection risk carried by UP. The immune modulatory effect of vitamin D has been extensively studied, and *in vitro* findings suggest that vitamin D deficiency could be a risk factor for infection [45]. Observational studies indicated that a low plasma vitamin D concentration before admission is associated with higher risks of respiratory tract infection, bloodstream infection or sepsis during hospitalization [46–50]. Despite the association discovered in observational studies, the evidence of vitamin D supplementation on preventing various infection is inadequate. The strongest evidence

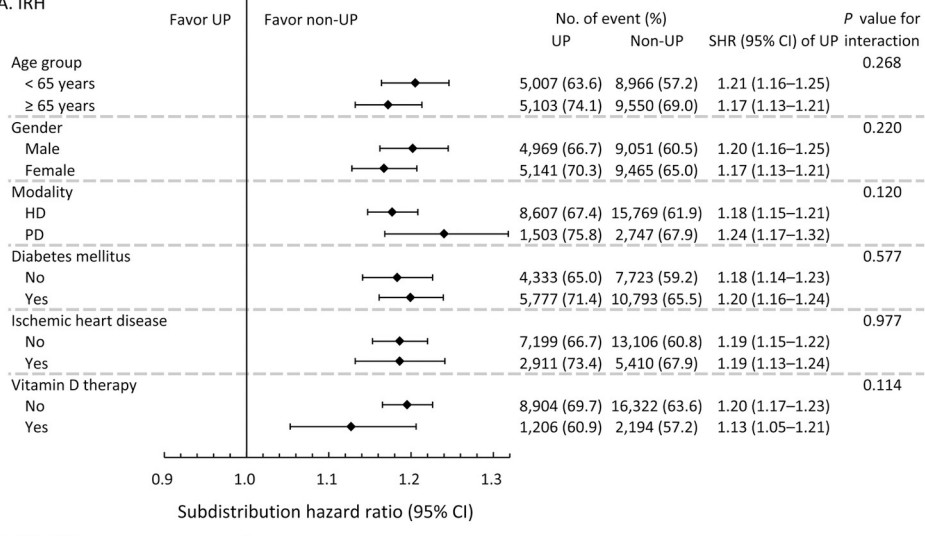

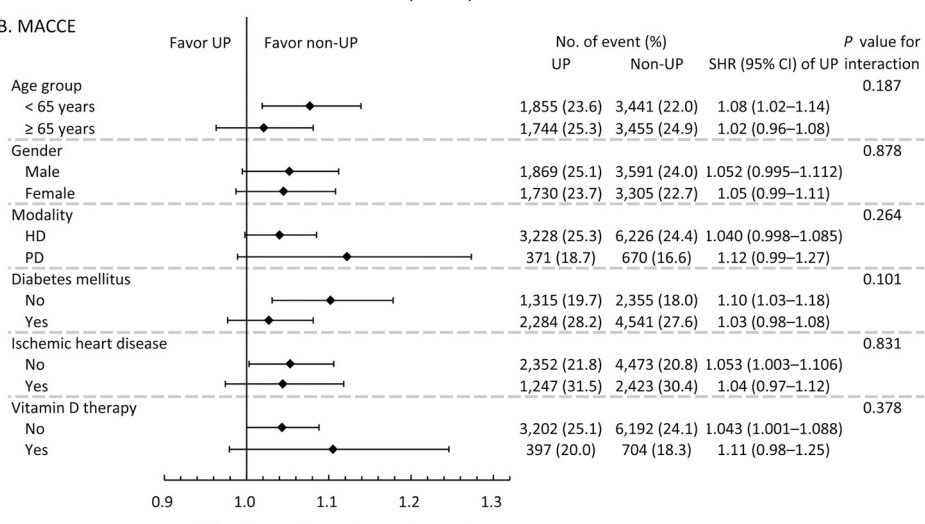

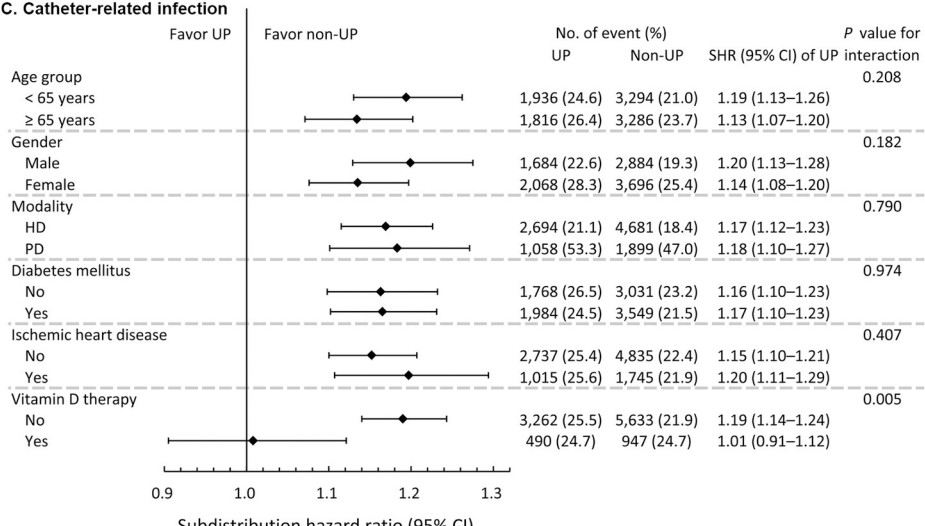

**Fig 3.** Prespecified subgroup analysis of outcomes, including infection-related hospitalization (A), major adverse cardiac and cerebrovascular event (B) and catheter-related infection (C).

supports the use of vitamin D in prevention of acute respiratory tract infection, especially in individuals who are vitamin D-deficient [51]. Whether vitamin D supplementation prevent the risk of other infections remains unclear and warrant further investigations.

Our study has certain limitations. First, the claim database does not contain information regarding pruritus severity. Some studies that have adopted the visual analogue scale (VAS) have demonstrated that severer pruritus was associated with poorer outcomes, whereas another study in Taiwan observed no such correlation [2–4, 6]. This inconsistency implies that a simple VAS may not be sufficient for predicting the overall effects of pruritus severity on outcomes. The Thai Renal Outcome Research group published a multidimensional assessment tool for UP, which included symptoms, signs, effects on sleep, and daily activities in 2017 [52]. This questionnaire may be adapted for future studies on UP. The lack of subjective responses from patients may also constitute a strength of our study because questionnaire-based study can be affected by interference as a result of recall bias or acquiescence bias [53, 54]. The second limitation of this study is the lack of lab data in the claims database. Our work may require further adjustment for identifying laboratory abnormalities or inadequate dialysis clearance. Third, we select UP patients by using the treatment criteria and did not classify disorders that may require long-term antihistamine or UVB therapy. Some diseases such as vitiligo (ICD 709.01) was extremely rare in our cohort (about 0.05%) and was consistent with previous study using Taiwan NHIRD [55]. The extremely low prevalence implies this disorder will not significantly affect our cohort selection by pruritus therapy. However, caution should be made when applying this criterion in population with higher prevalence of vitiligo.

## Conclusion

This nationwide population-based cohort study demonstrated a positive association between the presence of UP and increased risk of long-term morbidities, including infection-related hospitalization, MACCE, catheter-related infection and parathyroidectomy.

## Supporting information

**S1 Table. ICD-9 CM diagnostic codes.**
(DOCX)

**S1 Fig. Prespecified subgroup analysis of heart failure hospitalization.**
(DOCX)

**S2 Fig. Prespecified subgroup analysis of receiving parathyroidectomy.**
(DOCX)

## Acknowledgments

We are grateful to all members of the Taiwan Clinical Trial Consortium, TCTC. The authors also thank Alfred Hsing-Fen Lin and Zoe Ya-Jhu Syu for their assistance in statistical analysis.

## Author Contributions

**Conceptualization:** Sze-Wen Ting, Pei-Chun Fan, Chih-Hsiang Chang.

**Data curation:** Ming-Shyan Lin.

**Formal analysis:** Yu-Sheng Lin, Cheng-Chia Lee.

**Investigation:** Sze-Wen Ting, Pei-Chun Fan, Ming-Shyan Lin, Cheng-Chia Lee, George Kuo.

**Methodology:** Sze-Wen Ting, Pei-Chun Fan, Yu-Sheng Lin, Ming-Shyan Lin, George Kuo.

**Project administration:** Chih-Hsiang Chang.

**Software:** Yu-Sheng Lin.

**Supervision:** Chih-Hsiang Chang.

**Validation:** Cheng-Chia Lee, Chih-Hsiang Chang.

**Writing – original draft:** Sze-Wen Ting.

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
