## [Decision Letter · Decision Letter 0]

8 Jul 2020

PONE-D-20-17073

Uremic Pruritus and Non-Fatal Long-Term outcomes in the Dialysis Population

PLOS ONE

Dear Dr. Chang,

Thank you for submitting your manuscript to PLOS ONE. After careful consideration, we feel that it has merit but does not fully meet PLOS ONE’s publication criteria as it currently stands. Therefore, we invite you to submit a revised version of the manuscript that addresses the points raised during the review process.

Both reviewers raised several concerns, especially regarding definitions and data analysis. Particularly, the authors need to expand the size of their cohort, remove data related to fatal outcomes, and perform additional analysis on non-fatal outcomes to replace figure 3.

We look forward to receiving your revised manuscript.

Kind regards,

Yu Ru Kou, PhD

Academic Editor

PLOS ONE

Journal Requirements:

2. In the ethics statement in the manuscript, please provide additional information about the patient records used in your retrospective study. Specifically, please ensure that you have discussed whether all data/tissue samples  were fully anonymized before you accessed them and whether the IRB or ethics committee waived the requirement for informed consent.

4. Your ethics statement must appear in the Methods section of your manuscript. If your ethics statement is written in any section besides the Methods, please move it to the Methods section and delete it from any other section. Please also ensure that your ethics statement is included in your manuscript, as the ethics section of your online submission will not be published alongside your manuscript.

Reviewers' comments:

Reviewer's Responses to Questions

**Comments to the Author**

1. Is the manuscript technically sound, and do the data support the conclusions?

Reviewer #1: Partly

Reviewer #2: Yes

2. Has the statistical analysis been performed appropriately and rigorously? 

Reviewer #1: I Don't Know

Reviewer #2: Yes

3. Have the authors made all data underlying the findings in their manuscript fully available?

Reviewer #1: Yes

Reviewer #2: Yes

4. Is the manuscript presented in an intelligible fashion and written in standard English?

Reviewer #1: No

Reviewer #2: Yes

5. Review Comments to the Author

Reviewer #1: In the study, the association between uremic pruritus and non-fatal long term outcomes in dialysis patients were analyzed. The results showed that the risk of infection and cardiovascular events were increased. Here are the concerns from the reviewer:

1. The HR was only marginally increased (HR from 1 to around 1.1) which lessened the impact of this work.

2. What was the reason to exclude patients with parathyroidectomy?

3. For the outcomes, did the authors only count those with new events and exclude those with prior history, like what they did for parathyroidectomy?

4. What was the definition of infection-related hospitalization? There is a large spectrum of infection that requires hospitalization. Please try to classify the spectrum of infections, eg. viral/fungal/bacterial, common/opportunistic etc.

5. The definition of UP, "more than 42 daily doses of antihistamine", was a bit loose. How did the authors exclude pruritus caused by xerosis or xerotic dermatitis which is also a very common problem in dialysis patients.

6. What is the meaning of "non-fatal outcomes"? Infection and MACCE very often resulted in death, especially in immunocompromised patients. How did the authors tell "fatal" and "non-fatal" in this study?

Reviewer #2: Major commands

1. Your previous study in reference 13 discussed the association between uremic pruritus and long-term outcomes in patients undergoing dialysis. From the big database, it would be acceptable if the data presented were in sequence of a different sample size. I would strongly suggest to review the data presented.

2. You used standard deviation (STD) in Table 1 to represent the difference between 2 groups. It would be more clear to use p-value to show the results.

3. The information before matching does not need to be provided in table 1. It looked more complicated and I would suggest moving these to supplement data.

4. Your topic is about the association of uremic pruritis and non-fatal outcomes. It seems not suitable to put fatal outcomes in Table 2 as results.

5. In Figure 3, you did the subgroup analysis according to fatal outcomes. It would seem more suitable to do the analysis related to the non-fatal outcomes which is more related to your topic.

In general, the authors demonstrated expertise in big data analysis and in managing the data. Thus, I suggest taking into account to do several adjustments.

6. PLOS authors have the option to publish the peer review history of their article (what does this mean?). If published, this will include your full peer review and any attached files.

Reviewer #1: No

Reviewer #2: No

---

## [Author Response · Author response to Decision Letter 0]

21 Aug 2020

Response to reviewer 1

Reviewer #1: In the study, the association between uremic pruritus and non-fatal long-term outcomes in dialysis patients were analyzed. The results showed that the risk of infection and cardiovascular events were increased. Here are the concerns from the reviewer:

1. The HR was only marginally increased (HR from 1 to around 1.1) which lessened the impact of this work.

Thanks for your comment. We agreed that a marginally increased hazard ratio may imply that uremic pruritus contribute only a small portion to the adverse outcome in dialysis patients. However, we considered this result somehow reasonable, because other factors, such as traditional cardiovascular risk factors (diabetes, hypertension, dyslipidemia) and dialysis-specific risk factors, including hyperparathyroidism, deranged calcium-phosphate balance all have their effects on the morbidities of dialysis population. Uremic pruritus can be viewed as one comorbidity which is not negligible for predicting patient outcomes. 

2. What was the reason to exclude patients with parathyroidectomy?

Thanks for your comment. In some dialysis patients, an elevated parathyroid hormone with refractory pruritus could be the indication of parathyroidectomy, and some case series showed that parathyroidectomy may help relieve the extent of pruritus. We would like to observe this surgery during follow up, therefore, patients who underwent parathyroidectomy were excluded because they could not have this surgery again (if total parathyroidectomy) or the chance will decrease (if subtotal parathyroidectomy). 

3. For the outcomes, did the authors only count those with new events and exclude those with prior history, like what they did for parathyroidectomy?

For cardiovascular events, the disease burden may increase over time and episodes. We counted the events after index date, but used the past history for propensity score matching. For most of the infection, we counted the events after index date without considering the past history of infection. 

4. What was the definition of infection-related hospitalization? There is a large spectrum of infection that requires hospitalization. Please try to classify the spectrum of infections, eg. viral/fungal/bacterial, common/opportunistic etc.

We are sorry for we did not report the definition of infection-related hospitalization in the original manuscript. We used the discharged diagnoses of selected infection-related ICD-9 diagnostic codes in each emergency department visit or hospitalization. By using these ICD-9 codes, we could differentiate the sites of infection. However, the pathogen causing infection is unable to identify by using ICD-9 code. This part is addressed in the method (Definitions of Outcomes subsection) and limitation sections of revised manuscript. 

5. The definition of UP, "more than 42 daily doses of antihistamine", was a bit loose. How did the authors exclude pruritus caused by xerosis or xerotic dermatitis which is also a very common problem in dialysis patients.

Thanks for your comment. 

Xerosis is a common phenomenon in dialysis patients, and is sometimes viewed as one of the contributing/aggravating factors in uremic pruritus. If the itching improves a lot after application of emollient

We considered uremic pruritus as a “syndrome” rather a specific disease entity, therefore, we used a treatment-based criterion to select dialysis patients who has a considerable severity of itching. In previous studies, the diagnosis of uremic pruritus usually based on questionnaires or visual analogue scale, with the exclusion of primary cutaneous diseases that may require long-term use of systemic therapies, such as chronic urticaria and atopic dermatitis. The utilization of questionnaire was not diagnostic for uremic pruritus, but for classification of itching severity. 

During the study period, no mu-opioid receptor antagonist of kappa-opioid receptor agonist is available for treating uremic pruritus in Taiwan. Meanwhile, the prescription of gabapentinoids are not covered by national health insurance (NHI), therefore, the first line treatment for pruritus in dialysis patients are antihistamines. 

In the revised manuscript, we tried to sort out patients with more severe pruritus by increasing the threshold of antihistamine prescription to be equal or more than 84 days. Patients with any concomitant diagnoses that may require frequent antihistamine therapy are also excluded. The flowchart, all tables and figures and the text have been revised accordingly. 

6. What is the meaning of "non-fatal outcomes"? Infection and MACCE very often resulted in death, especially in immunocompromised patients. How did the authors tell "fatal" and "non-fatal" in this study?

Thanks for your comment. First, to avoid making confusion to the readers, we would like to adjust the term “non-fatal outcomes” into “outcomes” in the revised manuscript and “long-term morbidities” in the revised title. Second, as for the outcomes in our study, we only count those did not lead to death. That is to say, we count all emergency visit or hospitalization that patients were alive at discharge. The statistical analysis also utilized subdistribution hazard model, in which death is considered a competing risk. 

Response to reviewer 2

1. Your previous study in reference 13 discussed the association between uremic pruritus and long-term outcomes in patients undergoing dialysis. From the big database, it would be acceptable if the data presented were in sequence of a different sample size. I would strongly suggest to review the data presented.

Thanks for your comment. In correspondence with the other reviewer’s requirement, we changed to a stricter criterion of anti-histamine prescription to define uremic pruritus. The flowchart (and the number of eligible case), all tables and figures and the text have been revised accordingly. 

2. You used standard deviation (STD) in Table 1 to represent the difference between 2 groups. It would be more clear to use p-value to show the results.

We thank for your comment. We used “standardized difference” (STD) because of the huge sample size. The p-value tends to be extremely low when the sample size is huge. In this situation, using the standardized difference is more suitable because it is not affected by the size of sample.

3. The information before matching does not need to be provided in table 1. It looked more complicated and I would suggest moving these to supplement data.

Thanks for your opinion. We prefer to demonstrate the data before and after matching in the main text, because we think it is necessary to show the real-world characteristics (data before matching) of patients with and without uremic pruritus. Showing the data after matching is also necessary because the readers can easily assess the quality of matching.

4. Your topic is about the association of uremic pruritis and non-fatal outcomes. It seems not suitable to put fatal outcomes in Table 2 as results.

Thanks for your reminding. We removed all the term of “fatal outcome” in this revised manuscript. 

5. In Figure 3, you did the subgroup analysis according to fatal outcomes. It would seem more suitable to do the analysis related to the non-fatal outcomes which is more related to your topic.

We appreciate with your constructive suggestion. Based on the revised design, we performed subgroup analysis for selected morbidities/events which were significant between groups, including infection-related hospitalization, catheter related infection, major cardiovascular and cerebral composite adverse event, heart failure and parathyroidectomy. The Figure 3A-3B (forest plot for subgroup analysis) is revised and Supplemental Figure 1-3 have been added in this revised manuscript.

---

## [Decision Letter · Decision Letter 1]

7 Sep 2020

PONE-D-20-17073R1

Uremic Pruritus and Long-Term Morbidities in the Dialysis Population

PLOS ONE

Dear Dr. Chang,

Thank you for submitting your manuscript to PLOS ONE. After careful consideration, we feel that it has merit but does not fully meet PLOS ONE’s publication criteria as it currently stands. Therefore, we invite you to submit a revised version of the manuscript that addresses the points raised during the review process.

Both reviewers continued to have some concerns that require to be addressed.

We look forward to receiving your revised manuscript.

Kind regards,

Yu Ru Kou, PhD

Academic Editor

PLOS ONE

Reviewers' comments:

Reviewer's Responses to Questions

**Comments to the Author**

1. If the authors have adequately addressed your comments raised in a previous round of review and you feel that this manuscript is now acceptable for publication, you may indicate that here to bypass the “Comments to the Author” section, enter your conflict of interest statement in the “Confidential to Editor” section, and submit your "Accept" recommendation.

Reviewer #1: (No Response)

Reviewer #2: (No Response)

2. Is the manuscript technically sound, and do the data support the conclusions?

Reviewer #1: Yes

Reviewer #2: Yes

3. Has the statistical analysis been performed appropriately and rigorously? 

Reviewer #1: Yes

Reviewer #2: Yes

4. Have the authors made all data underlying the findings in their manuscript fully available?

Reviewer #1: Yes

Reviewer #2: Yes

5. Is the manuscript presented in an intelligible fashion and written in standard English?

Reviewer #1: Yes

Reviewer #2: Yes

6. Review Comments to the Author

Reviewer #1: The current version showed substantial improvement. Some concerns remained from the reviewer:

1. The study populations is still a concern although the authors made efforts to re-define the populations. Did the authors exclude those who already started using antihistamine prior to the index date in the UP group? UVB treatment is frequently applied on psoriasis and vitiligo patients. Psoriasis was excluded as UP group. How about vitiligo?

2. Please recheck the panels in figure 3 and supplementary figure 1. The panels were mistaken here. Catheter-related infection should be placed in the main manuscript rather than in the supplementary data since it was a focus of discussion in the main text.

3. Supplementary Table 1: "Urticarial" should be "urticaria".

Reviewer #2: Dear Authors,

All my commands have been addressed adequately.

I made the following suggestions:

1. Please include the complete information of the reference 13. Accepted, unpublished article would be cited same as published articles, but substitute “Forthcoming” for page numbers or DOI.

2. I understand the author tried to match all factors that may affect the outcomes; however, some of the factors such as the dialysis duration was not included in the factor of matching. I suggested not include the information before matching because it looked more complicated.

In general, the authors tried to revise as the commands.

Thank you very much for your attention to my opinion.

7. PLOS authors have the option to publish the peer review history of their article (what does this mean?). If published, this will include your full peer review and any attached files.

Reviewer #1: No

Reviewer #2: No

---

## [Author Response · Author response to Decision Letter 1]

23 Sep 2020

Reviewer #1: The current version showed substantial improvement. Some concerns remained from the reviewer:

1. The study populations is still a concern although the authors made efforts to re-define the populations. Did the authors exclude those who already started using antihistamine prior to the index date in the UP group? UVB treatment is frequently applied on psoriasis and vitiligo patients. Psoriasis was excluded as UP group. How about vitiligo?

Thanks for your comment. We agreed that the current flowchart may count patients with vitiligo who also receive UVB therapy. The patients diagnosed with vitiligo were identified with the ICD-9 code 709.01. Within 52,357 patients who have any doses of antihistamine prescription or UVB therapy, only 24 patients were found to have vitiligo (24 / 52357 = 0.05%). This low prevalence of vitiligo is compatible with Chen et al., who also used Taiwan NHIRD and found a prevalence around 0.064% in general population of Taiwan (Comorbidity profiles in association with vitiligo: a nationwide population-based study in Taiwan. Chen YT, Chen YJ, Hwang CY, et al. J Eur Acad Dermatol Venereol. 2015 Jul;29(7):1362-9. doi: 10.1111/jdv.12870.). Because of the extremely low prevalence of vitiligo in our advanced CKD & ESRD cohort, we think it may barely affect the events of our study. We have written this limitation into the limitation section. 

We did not exclude patients who use antihistamine before index date, because uremic pruritus is also very frequent at the stage of advanced chronic kidney disease, thus, they may use antihistamines before they initiate dialysis. Some patients’ pruritus improves after starting dialysis therapy; however, others may continue to have itching despite the dialysis. Hence, the main interest in this study focus on the long-term impact of uremic pruritus which still exist after starting dialysis. We addressed this focus in the revised manuscript.

2. Please recheck the panels in figure 3 and supplementary figure 1. The panels were mistaken here. Catheter-related infection should be placed in the main manuscript rather than in the supplementary data since it was a focus of discussion in the main text.

Thanks for your suggestion. We have put supplementary fig 1 and figure 3A-B together (revised to be figure 3A-C)

3. Supplementary Table 1: "Urticarial" should be "urticaria".

Thank you very much. We made the correction in the revised supplemental document.

Reviewer #2: 

Dear Authors,

All my commands have been addressed adequately.

I made the following suggestions:

1. Please include the complete information of the reference 13. Accepted, unpublished article would be cited same as published articles, but substitute “Forthcoming” for page numbers or DOI.

Thank you very much. This research letter has been published in July 2020. We complete the information (including journal volume, issue and page) of this reference in the revised manuscript. 

2. I understand the author tried to match all factors that may affect the outcomes; however, some of the factors such as the dialysis duration was not included in the factor of matching. I suggested not include the information before matching because it looked more complicated.

Thanks for your comment. The variables listed in Table 1 were all included in the calculation of propensity score, where the “follow up duration” indicated the duration from index date to the end of follow up. We did not include dialysis vintage in table 1, because we start the follow up at the first day that patients start dialysis; therefore, at baseline, all patients’ dialysis vintage is zero. Based on the above reason, we tend to retain data of both the before and after matching.

---

## [Decision Letter · Decision Letter 2]

6 Oct 2020

PONE-D-20-17073R2

Uremic Pruritus and Long-Term Morbidities in the Dialysis Population

PLOS ONE

Dear Dr. Chang,

Thank you for submitting your manuscript to PLOS ONE. After careful consideration, we feel that it has merit but does not fully meet PLOS ONE’s publication criteria as it currently stands. Therefore, we invite you to submit a revised version of the manuscript that addresses the points raised during the review process.

The reviewer #1 pointed out a minor issue regarding the legend of figure 3. Please fix the problem

We look forward to receiving your revised manuscript.

Kind regards,

Yu Ru Kou, PhD

Academic Editor

PLOS ONE

Reviewers' comments:

Reviewer's Responses to Questions

**Comments to the Author**

1. If the authors have adequately addressed your comments raised in a previous round of review and you feel that this manuscript is now acceptable for publication, you may indicate that here to bypass the “Comments to the Author” section, enter your conflict of interest statement in the “Confidential to Editor” section, and submit your "Accept" recommendation.

Reviewer #1: (No Response)

Reviewer #2: All comments have been addressed

2. Is the manuscript technically sound, and do the data support the conclusions?

Reviewer #1: Yes

Reviewer #2: Yes

3. Has the statistical analysis been performed appropriately and rigorously? 

Reviewer #1: Yes

Reviewer #2: Yes

4. Have the authors made all data underlying the findings in their manuscript fully available?

Reviewer #1: No

Reviewer #2: Yes

5. Is the manuscript presented in an intelligible fashion and written in standard English?

Reviewer #1: Yes

Reviewer #2: Yes

6. Review Comments to the Author

Reviewer #1: The questions were addressed properly. There is one minor point. The legend of figure 3 was not revised according to the change of the panels.

Reviewer #2: Dear Authors,

All my commands have been addressed adequately.

In general, the authors tried to revise as the commands.

Thanks for your attention to my opinion.

7. PLOS authors have the option to publish the peer review history of their article (what does this mean?). If published, this will include your full peer review and any attached files.

Reviewer #1: No

Reviewer #2: **Yes: **Lai, Yu-Hsien

---

## [Author Response · Author response to Decision Letter 2]

6 Oct 2020

Response to the reviewers

Reviewer #1: The questions were addressed properly. There is one minor point. The legend of figure 3 was not revised according to the change of the panels.:

Thank you very much. We revised the figure legend of figure 3.

---

## [Editor Report · Decision Letter 3]

8 Oct 2020

Uremic Pruritus and Long-Term Morbidities in the Dialysis Population

PONE-D-20-17073R3

Dear Dr. Chang,

We’re pleased to inform you that your manuscript has been judged scientifically suitable for publication and will be formally accepted for publication once it meets all outstanding technical requirements.

Kind regards,

Yu Ru Kou, PhD

Academic Editor

PLOS ONE
---

## [Editor Report · Acceptance letter]

16 Oct 2020

PONE-D-20-17073R3 

Uremic Pruritus and Long-Term Morbidities in the Dialysis Population 

Dear Dr. Chang:

I'm pleased to inform you that your manuscript has been deemed suitable for publication in PLOS ONE. Congratulations! Your manuscript is now with our production department. 

Kind regards, 

on behalf of

Dr. Yu Ru Kou 

Academic Editor

PLOS ONE